# Validation of a Food Frequency Questionnaire to Assess Intake of *n-3* Polyunsaturated Fatty Acids in Switzerland

**DOI:** 10.3390/nu11081863

**Published:** 2019-08-10

**Authors:** Isabelle Herter-Aeberli, Celeste Graf, Anna Vollenweider, Isabelle Häberling, Pakeerathan Srikanthan, Martin Hersberger, Gregor Berger, Déborah Mathis

**Affiliations:** 1Laboratory of Human Nutrition, Institute of Food, Nutrition and Health, ETH Zurich, 8092 Zurich, Switzerland; 2Department of Child and Adolescent Psychiatry, University Hospital of Psychiatry Zurich, 8032 Zurich, Switzerland; 3Clinical Chemistry and Biochemistry, University Children’s Hospital Zurich, 8032 Zurich, Switzerland

**Keywords:** food frequency questionnaire, *n-3* PUFA, polyunsaturated fatty acids, food record, validation, dietary intake

## Abstract

Population-based data suggest that high intake of omega-3 (*n-3)* polyunsaturated fatty acids (PUFA) may be beneficial in a variety of health conditions. It is likely that mainly those patients with preexisting *n-3* deficiency are those that benefit most from *n-3* fatty acid supplementation. Therefore, for targeted interventions, a fast and reliable screening tool for *n-3* PUFA intake is necessary. Thus, the aim of this project was to adapt and validate a food frequency questionnaire (FFQ) for *n-3* PUFA intake in Switzerland while using as references the following: (1) 7-day food records (FR), and (2) *n-3* fatty acid composition of red blood cells (RBC). We recruited 46 healthy adults for the first part of the study and 152 for the second. We used the dietary software EBISpro for the analysis of *n-3* PUFA intake. RBC fatty acid composition was determined by gas chromatography mass spectrometry (GC-MS). Using correlation analysis, we found a moderate significant association between FFQ and FR for α-linolenic acid (ALA), eicosapentanoic acid (EPA), docosahexanoic acid (DHA), and total *n-3* fatty acids (all *r* between 0.523 and 0.586, all *p* < 0.001). Bland Altman analysis further showed good agreement between the two methods and no proportional bias. Correlations between FFQ and RBC fatty acid composition were also moderate for EPA and DHA (*r* = 0.430 and *r* = 0.605, *p* < 0.001), but weaker for ALA and total *n-3* (*r* = 0.314 and *r* = 0.211, *p* < 0.01). The efficacy of the FFQ to classify individuals into the same or adjacent quartile of RBC PUFA content ranged between 70% and 87% for the different fatty acids. In conclusion, we showed that the Swiss *n-3* PUFA FFQ is a valid tool to assess dietary *n-3* PUFA intake, especially DHA and EPA, to determine population groups at risk for low intake.

## 1. Introduction

The first evidence of a beneficial effect of omega-3 (*n-3*) polyunsaturated fatty acids (PUFA) on human health was described based on population data from Greenland Eskimos between 1950 and 1974. Eskimos traditionally consume a diet very high in seafood and were found to have a low prevalence of several chronic diseases, including coronary heart disease, asthma, diabetes mellitus and multiple sclerosis [1]. Since those early observations, sufficient PUFA intake has been widely recognized as an important determinant of a variety of health conditions, including inflammatory conditions, mental health, cardiovascular disease, and cancer [2,3,4,5,6,7]. Over the centuries, but especially since the industrial revolution, the western diet has moved from a balanced omega-6 *(n-6)* to *n-3* ratio towards a major excess in *n-6* fatty acids. A relative deficiency in *n-3* PUFAs with an estimated ratio of *n-6* to *n-3* PUFA of 15-20/1, as opposed to 1/1, has been the result of this change in dietary patters [8]. Numerous researchers propose that the increase in civilization diseases such as diabetes, metabolic syndrome, but also mental disorders such as depression and attention deficit and hyperactivity disorders may also be related to this shift towards a more “proinflammatory diet” [9].

With the growing burden of chronic, non-communicable disease in western societies, the importance of understanding the different influencing factors increases. In order to study intervention effects and relate intake to disease risk, however, having a rapid and reliable screening tool for *n-3* PUFA status is essential. The best source to assess long-term fatty acid intake is considered to be adipose tissue fatty acid composition, while red blood cell (RBC) fatty acid composition reflects intakes over the past 120 days and plasma fatty acid composition is an indication for intakes over the past few days [10]. All of these parameters require more or less invasive tissue sampling and are thus limited to clinical settings. Dietary assessments, on the other hand, allow us to approximate the nutrient intake in a completely non-invasive way, making the methods suitable for large population groups in all settings. As *n-3* PUFAs are largely exogenic, meaning they cannot be synthesized de novo by the human body, the major sources are intake via diet or supplements. Correlational analyses between dietary assessments and biomarkers yield good results [10,11,12]. Different methods exist for dietary assessments, and each has its advantages and disadvantages. The weighed food record is considered the gold standard in dietary assessments, but its application imposes a relatively high burden on the subjects. Furthermore, there is a risk that subjects change their diet during the recording period to simplify recording or to better comply with a perceived healthy diet. A food frequency questionnaire (FFQ) is generally used to assess the intake of specific foods or food groups, but can also be used for the assessment of individual nutrients with certain adaptations. The foods that contribute to *n-3* fatty acid intake in the human diet are relatively limited, and the use of an FFQ is therefore adequate in this case. The advantages of an FFQ are that the burden on the subjects is low and that it is therefore possible to include a large sample size. Furthermore, as it is a retrospective method, there is no risk of altered food habits for the recording period [13].

The aim of this study was to adapt and validate an existing FFQ for *n-3* PUFA intake from the United States for use in Switzerland using RBC *n-3* fatty acid composition and 7-day food records (FR) as references.

## 2. Methods

### 2.1. Questionnaire

We have adapted a self-report questionnaire based on the American *n-3* PUFA FFQ by Sublette et al. which was developed and validated to assess *n-3* PUFA intake in the New York City area [14]. The adaptation was done based on dietary habits, as well as sales figures for fish and seafood in Switzerland from the years 2014–2016. The main adaptations were in the choice of fish and seafood products in the questionnaire, as well as the addition of plant-based margarine and chia seeds. The questionnaire asked about consumption frequencies over the past 6 months. The final Swiss *n-3* PUFA FFQ is available as online Appendix A.

### 2.2. Study Population

The study was conducted in two steps: (1) validation of the FFQ against 7-day FR, and (2) validation of the FFQ against RBC *n-3* PUFA composition. All participants were recruited from the student and staff population of ETH Zurich (both parts), staff of the University Hospital of Psychiatry Zurich (part 1) as well as staff of the Children’s Hospital Zurich (part 2). The study was approved by the Institutional Ethics Committee of the state of Zurich (BASEC Nr. 2017-01370) and was registered under clinicaltrials.gov (NCT03409445).

For part 1, we recruited 46 healthy male and female participants aged between 19 and 53 years and willing to fill in a 7-day weighed FR and the adapted Swiss *n-3* PUFA FFQ. No health-related data from the participants were assessed. For part 2 we recruited 152 healthy male and female participants aged between 18 and 59 years. Participants were excluded if they suffered from a self-reported chronic disease of the gastrointestinal tract that might influence lipid absorption. For women, pregnancy and lactation were further exclusion criteria.

### 2.3. Study Design

For part 1, interested volunteers met with the study assistant to receive detailed instructions for completion of the 7-day weighed FR. They were asked to record all food and drink consumed over a period of 7 consecutive days as precisely as possible. They were asked to weigh the foods consumed whenever possible and were provided with a kitchen scale if needed. If weighing was not possible for a certain meal, participants were asked to estimate portion sizes using the picture book from the National Nutrition Survey menuCH [15]. Whenever possible, individual ingredients should be reported rather than final meals, including preparation methods, brand names, added oil, etc. Special emphasis was placed on not forgetting snacks between meals, sauces or salad dressings, and to mention fat content of all products (e.g., dairy products). The importance of not changing eating habits during the 7 days of recording was also stressed. After receiving the instructions for the 7-day weighed FR, the participants were asked to complete the Swiss *n-3* PUFA FFQ directly at the research institute. When they returned the completed FR, the study assistant reviewed all entries with the participants to rule out potential misunderstandings and misreporting. Data was collected between March and May 2017, and between February and March 2018.

For part 2 interested volunteers received the participant information sheet by e-mail. Those who were still interested in participating were invited to the Human Nutrition Laboratory of the ETH Zurich. The study procedure was again explained, and potential questions were answered before obtaining informed written consent. A venous blood sample of 4 mL was then collected into EDTA coated vacutainers and weight and height of the participants was measured. After blood sampling, the participants were asked to complete the Swiss *n-3* PUFA FFQ directly at the research institute. All samples and data were collected between January and March 2018.

### 2.4. Dietary Record Analysis

Both the 7-day FR and the Swiss *n-3* PUFA FFQs were analyzed using the nutrition software EBISpro for Windows 2011 (Dr. J. Erhart, University of Hohenheim, Stuttgart , Germany). The *n-3* PUFAs of interest were total *n-3* PUFAs, ALA, EPA, and DHA. Total *n-3* PUFAs were calculated as the sum of ALA + EPA + DHA + docosapentanoic acid (DPA). DPA was included as part of total *n-3* PUFA, even though its individual intake was not investigated. Missing data for *n-3* PUFA content of the products was added either from Food Composition and Nutrition Tables by Souci, Fachmann, Kraut [16] or from the USDA Food Composition Database [17]. The foods listed in the Swiss *n-3* PUFA FFQ were entered as a template in the software. For the analysis, only the frequency of consumption indicated by the participants had to be entered for each product. If the participants indicated having eaten several different species of fish, the recorded amount and frequency was divided by the number of species consumed. As very few participants who consumed *n-3* PUFA containing dietary supplements could report the exact product name and amounts, we decided not to include the amount of *n-3* PUFAs consumed from supplements in the analysis but divided the participants by supplement consumers (*n* = 14) and non-consumers.

For the analysis of the 7-day weighed FR, each product recorded was searched in the database and the indicated amount entered. The intake was averaged over the 7 days and this mean value was used for data analysis for the different *n-3* PUFAs.

### 2.5. Fatty Acids Analysis in RBC

Blood samples for fatty acid analysis in RBC were collected into EDTA coated vacutainers and plasma was separated by centrifugation at 2000 *g* for 10 min at room temperature within 2 h of collection. The samples containing the RBC were stored at 4 °C for a maximum of seven days until further processing. The RBC were washed with 0.9%-NaCl solution and stored at −80 °C in Eppendorf tubes treated with 1%-BHT solution until analysis. The determination of fatty acids in RBC was carried out by gas chromatography coupled to a tandem mass spectrometer (GC-MS/MS, Thermo Scientific TSQ 8000, Waltham, MA, USA), according to the method of Moser et al. [18,19]. Briefly, the fatty acids in the RBC were extracted with a mixture of MeOH and Dichloromethane, and the organic phase was derivatized with acetyl chloride. The resulting methyl ester fatty acids were purified by liquid-liquid extraction with hexane. The samples were then injected into the GC-MS/MS system and recorded using selective reaction monitoring (SRM). The quantity of each fatty acid was expressed as a percentage of total fatty acids in RBC. The Omega-3 Index was calculated as the proportion of EPA + DHA as a percentage of total RBC fatty acids.

### 2.6. Data Analysis

Data analysis was done using IBM SPSS Statistics Version 24. All data was checked for normality using the Shapiro Wilks test. As neither the *n-3* PUFA intake nor all RBC fatty acid concentrations were normally distributed, data analysis was done using non-parametric tests. The Wilcoxon signed-rank test was used to compare median intakes assessed by the 7-day FR and the FFQ. We further evaluated the strength and direction of the relationship between the two methods using Spearman’s correlation coefficients. In addition, we used Bland-Altman plots to visualize the agreement across the range of intake. To do this, we plotted the difference between the two methods against their respective average. The limits of agreement were determined as ±1.96 SD for each fatty acid. One sample t-tests on the difference were used to test whether the mean difference is constant across the range of means. To check for proportional bias, linear regression analysis was used with the difference as the dependent and the mean as the independent variable. To investigate the strength and direction of the relationship between *n-3* PUFA intake assessed using the Swiss *n-3* PUFA FFQ and RBC PUFA concentrations, we used Spearman’s correlation coefficients. To assess the importance of other factors, we used multiple linear regressions with the different RBC PUFAs as the dependent variable, PUFA intake from the Swiss *n-3* PUFA FFQ, as well as gender, age, supplement intake, and fish consumption as the independent variables. Differences between RBC fatty acid compositions of different groups (male vs. female, supplement users vs. non-users) were assessed using the Mann-Whitney U-test. Participants were classified into quartiles based on Swiss *n-3* PUFA FFQ-estimated fatty acid intake and plasma concentrations. Efficacy of the FFQ to place participants in the same or adjacent quartile was further calculated, as well as the misclassification into extreme quartiles.

## 3. Results

Characteristics of both study populations are shown in Table 1. Both groups were comparable with regard to age and included more women than men.

### 3.1. Questionnaire Validity Based on FR

Median *n-3* PUFA intake assessed by the FR and the Swiss *n-3* PUFA FFQ, as well as results of the correlation analysis and the group comparisons, are shown in Table 2. Neither total *n-3* PUFA intake nor the individual PUFAs differed significantly between the assessment by FR and FFQ. Further, we determined significant correlations with a Spearman correlation coefficient between 0.5 and 0.6 between FR and FFQ for all PUFA’s investigated. The Bland Altman plots visualizing the agreement between the two methods for ALA, EPA, DHA, and total *n-3* PUFA are shown in Figure 1. For all plots, the mean difference lies close to zero and most points are within the limits of agreement of the mean ±1.96 SD. The one sample t-test of the differences was non-significant for all individual PUFAs and for total *n-3* PUFA intake (ALA: *p* = 0.929; EPA: *p* = 0.922; DHA: *p* = 0.609; total *n-3*: *p* = 0.886). Furthermore, the linear regressions produced unstandardized B values close to 0 (ALA: B = 0.020; EPA: B = −0.260; DHA: B = −0.049, and total *n-3*: B = 0.007) with non-significant *p*-values (ALA: *p* = 0.880; EPA: *p* = 0.192; DHA: *p* = 0.718, and total *n-3*: *p* = 0.957), indicating no proportional bias.

### 3.2. Questionnaire Validity Based on RBC PUFA Composition

Median PUFA intake assessed by the Swiss *n-3* PUFA FFQ, RBC PUFA concentrations, and the results of the correlation analyses are shown in Table 3. We found strongly significant correlations between intake as assessed by Swiss *n-3* PUFA FFQ and RBC PUFA composition, especially for EPA (*r* = 0.430) and DHA (*r* = 0.605), and weaker, but still significant correlations for ALA (*r* = 0.314) and total *n-3* PUFAs (*r* = 0.211). We used linear regression analysis to identify further predictors of RBC *n-3* PUFA composition using gender, age, supplement intake, and fish consumption as independent variables besides FFQ PUFA. For all four analyses (ALA, EPA, DHA, and total *n-3*) the best model included all four additional factors, even though they did not all show a significant effect (compare Table 4). Equations (1)–(4) can be used to estimate the different RBC PUFA proportions based on FFQ data of the respective PUFA, gender, age, supplement intake (yes/no), and fish intake (yes/no):RBC ALA (%) = 0.102 + (0.013 × calculated FFQ ALA) − (0.014 × gender) − (0.004 × supplement intake) + (0.004 × fish intake)(1)
RBC EPA (%) = 0.368 + (1.497 × calculated FFQ EPA) − (0.016 × gender) + (0.005 × age) + (0.280 × supplement intake) − (0.053 × fish intake)(2)
RBC DHA (%) = 5.344 + (5.029 × calculated FFQ DHA) − (0.631 × gender) + (0.013 × age) + (0.595 × supplement intake) − (1.397 × fish intake)(3)
RBC total *n-3* (%) = 7.158 + (0.246 × calculated FFQ total *n-3*) − (0.323 × gender) + (0.021 × age) + (1.612 × supplement intake) − (1.874 × fish intake)(4)
Equations (1)–(4): Estimation of RBC *n-3* PUFA composition based on *n-3* PUFA intake estimated using the FFQ. Gender: 1 = male, 0 = female; supplement intake: 1 = *n-3* supplement intake, 0 = no supplement; fish consumption: 1 = no fish consumption, 0 = fish consumption; Age was omitted in Equation (1) as the coefficient was 0.

The proportion of RBC PUFAs differed between males and females for ALA (*p* < 0.001) and EPA (*p* = 0.029), but was not significant for DHA, total *n-3* PUFA, *n-3* index, or n-6/*n-3* ratio (all *p* > 0.05). The proportion of ALA in RBC was higher in females, while the proportion of EPA was higher in males. PUFA intake assessed by FFQ differed between males and females for EPA (*p* = 0.009) and DHA (*p* = 0.006) and was not different for ALA and total *n-3* (both *p* > 0.05). Again, EPA intake was higher in males, and so was DHA intake.

The proportion of RBC PUFA differed significantly between supplement consumers and non-consumers for all components assessed (ALA: *p* = 0.018, EPA, DHA, total *n-3* PUFA, *n-3* index, n-6/*n-3* ratio: *p* < 0.001), with higher intakes and a lower n-6/*n-3* ratio for supplement consumers. The intake assessed by the Swiss *n-3* PUFA FFQ differed between supplement consumers and non-consumers for EPA (*p* < 0.001) and DHA (*p* < 0.001), but not for ALA and total *n-3* intake (*p* > 0.05). Intakes were higher in supplement consumers compared to non-consumers.

The efficacy to classify individuals into the same or adjacent quartile of RBC PUFA content was 73.0% for ALA, 78.3% for EPA, 87.5% for DHA, and 70.4% for total *n-3*. Misclassification into extreme quartiles amounted to 17.8% for ALA, 13.8% for EPA, 7.9% for DHA, and 22.4% for total *n-3*.

## 4. Discussion

We have developed an adapted Swiss *n-3* PUFA FFQ to approximate the intake of *n-3* PUFA intake in Swiss populations based on an existing, validated questionnaire developed in the US [14]. As FFQs are sensitive to different dietary practices, it is important to validate a new questionnaire in the target population. For a more objective validation of the Swiss *n-3* PUFA FFQ, we chose to use not only the gold standard in dietary assessment, 7-day weighed food records, but also biomarkers of PUFA status (RBC PUFA content). The two validations were done in two separate groups of volunteers. We have been able to confirm good relative validity of the Swiss *n-3* PUFA FFQ using both approaches.

The strength of the correlations found between FFQ and FR was moderate (between *r* = 0.523 and *r* = 0.586) and similar to a previous validation study for an FFQ for *n-3* PUFA intake in Australian children (between *r* = 0.684 and *r* = 0.691) [20]. The range of intake assessed using the Swiss *n-3* PUFA FFQ and FR was comparable, although the FFQ tended to estimate higher intake for EPA and DHA. This can be explained by the fact that the Swiss *n-3* PUFA FFQ focuses on fish/seafood, as well as a few selected plant-based products containing high amounts of *n-3* PUFA. While the main dietary sources for EPA and DHA in the human diet are fish and seafood, ALA sources (and consequently total *n-3* PUFA sources) also include a variety of oils and other plant-based foods, which were not all represented in the Swiss *n-3* PUFA FFQ. Thus, based on the recall period of six months, we may have a slightly skewed estimation of EPA and DHA using the Swiss *n-3* PUFA FFQ, as some study participants may not have consumed fish or seafood in the 7-day FR period, but did within the past six months. On the other hand, the Swiss *n-3* PUFA FFQ may not have captured all ALA and total *n-3* PUFA sources, thus leading to a slight underestimation of those groups of food. Nevertheless, the median intakes did not differ significantly between the two methods for any of the PUFAs reported. It was reported previously that in general FFQs tend to overestimate dietary intake compared to other methods such as repeated 24-h recalls or food records [21,22,23]. In our current study, the differences were not significant. Using Bland-Altman plots to compare the results from the Swiss *n-3* PUFA FFQ with FR, we found mean differences close to zero for all PUFAs investigated, indicating no systematic bias. As most of the points were further found to be within the calculated limits of agreement, those plots support the validity of our questionnaire. Nevertheless, it needs to be considered that the limits of agreement in the Bland-Altman plots are in the range of the mean values and are therefore rather large. Thus, the agreement between the two methods should be treated cautiously, even though there is no systematic bias.

The strength of the correlations between *n-3* PUFA intake assessed using the Swiss *n-3* PUFA FFQ and RBC *n-3* PUFA composition was more variable compared to the above comparison between the Swiss *n-3* PUFA FFQ and FR, and ranged from *r* = 0.211 for total *n-3* to *r* = 0.605 for DHA. The correlation coefficients for ALA, EPA, and DHA found in the present study (*r* = 0.314, *r* = 0.430 and *r* = 0.605) were a little higher, but overall in a similar range as those found by Sublette et al. in their validation (*r* = 0.22, *r* = 0.38, and *r* = 0.50 for ALA, EPA, and DHA, respectively) [14]. However, the study population in the study by Sublette et al. was potentially more diverse, as they included both healthy participants and patients with major depressive disorder [14]. We found the strongest correlation for DHA, suggesting the best prediction of RBC DHA content by the Swiss *n-3* PUFA FFQ. The correlation found for EPA can still be considered good, if considerably lower than the one for DHA. However, in the case of EPA, it also has to be considered that physiological processes may influence the relationship between *n-3* EPA intake and RBC EPA composition. One of those factors is the low conversion rate of ALA into EPA, which is strongly influenced by the inhibitory effects of *n-6* PUFA intake [24]. The Swiss *n-3* PUFA FFQ does not assess *n-6* PUFA intake, as this would have been much more complex and time-consuming due to the many food items containing *n-6* PUFAs. Thus, we were not able to control for this potential confounding effect. Furthermore, polymorphisms in the Δ-5 and Δ-6 fatty acid desaturase genes have been identified, which were shown to affect variations in EPA but not DHA [25]. Again, the fact that the weakest correlations were found for ALA and total *n-3* PUFA is an indication that the Swiss *n-3* PUFA FFQ did not assess all sources of ALA. This agrees with other studies that also found weak or even non-significant correlations for ALA [26,27,28]. A further indication is that ALA supplementation studies have found only small effects on plasma phospholipid ALA content, which can be explained by several factors. Firstly, ALA can be converted to EPA and eventually DHA through elongases and desaturases, but this conversion is highly variable. Secondly, a significant amount of ALA is suggested to be oxidized for energy production [12].

We found significant gender differences in both RBC PUFA composition and PUFA intake. While RBC ALA was significantly higher in females, ALA intake did not differ between sexes. Even though this seems controversial at first sight, the agreement between the two methods to assess ALA was only limited, as not all ALA sources were assessed within the questionnaire. Thus, women may have consumed higher amounts of the non-detected ALA compared to men, which would explain the discrepancy in the findings. On the other hand, DHA intake was found to be higher in males, while there was no significant difference for RBC DHA content. This may be related to the more efficient conversion of ALA to DHA by women (up to 9%) compared to men (<1%) [12,29,30]. In women, a higher proportion of DHA may have been converted from ALA rather than taken in through the diet directly.

The Swiss *n-3* PUFA FFQ validated here is intended for use as a screening tool to identify individuals with low *n-3* PUFA intakes in order to potentially improve their diet and select high risk groups with a low omega-3 status. It was thus important to investigate whether the tool can accurately identify consumers with low intakes. Using a quartile analysis, we have shown that the Swiss *n-3* PUFA FFQ places participants into the same or adjacent quartile with an accuracy between 70% and 87% for the different fatty acids. The highest accuracy was found for DHA, followed by EPA, in agreement with the strength of correlations discussed above. These findings are in agreement with the study by Sublette et al. where the highest efficacy to accurately place participants in the same or adjacent quartile was with 83% also found for DHA [14]. We found the lowest rate of misclassification into the extreme quartile for DHA with 7.9% and the highest for total *n-3* PUFAs with 22.4%. This is again in agreement with the explanation above that the FFQ was better able to capture DHA and EPA intakes rather than ALA and total *n-3*. Furthermore, total *n-3* PUFA contain not only ALA, EPA, and DHA, but also DPA, which occurs in foods in much smaller amounts than the other three *n-3* PUFAs. RBC or plasma DPA content is mainly derived from endogenous elongation of EPA and can also undergo retroconversion back to EPA, and thus is not expected to correlate well with dietary intake of DPA [31]. This fact is likely to lead to a blunted agreement between RBC total *n-3* PUFA and FFQ total *n-3* PUFA. Despite the relatively high efficacy in placing participants in the same or adjacent quartiles, especially for DHA and EPA, misclassification does happen. Therefore, in order to have an accurate diagnosis of low *n-3* PUFA status, a biochemical analysis should be favored whenever possible. However, the presented FFQ can be a valid screening tool, especially in population-based studies.

Different recommendations exist for the intake of *n-3* PUFAs. On one hand, the International Society for the Study of Fatty Acids and Lipids (ISSFAL) defined a minimum intake of total *n-3* PUFA at 500 mg/day [32]. The American Heart Association Nutrition Committee (AHANC) recommends a minimum of 300 mg/day of total *n-3* PUFA while the European Food Safety Authority (EFSA) recommends a minimum EPA + DHA intake of 250 mg/day [33]. To our knowledge, no recommendations are available for high risk populations, such as patients with dyslimidemia, cardiovascular disorders or major mental disorders. Such high-risk populations may need much larger amounts of *n-3* PUFAs than non-affected healthy controls. In the validation of the Swiss *n-3* PUFA FFQ against FR, total *n-3* PUFA intake assessed by both methods was above both the ISSFAL and the AHANC recommendations, while in the validation of the FFQ against RBC PUFAs, *n-3* PUFA assessed by the FFQ was slightly below the ISSFAL but above the AHANC recommendation. However, in none of the three assessments of dietary intake, the EFSA recommendation for EPA + DHA intake was reached. In general, intakes observed in our study were lower than what was found in most other European countries [34]. The median omega-3 index calculated for our study population was 6.1%, which lies below the desirable target level for protection of coronary heart disease of >8% [35]. Thus, it seems that our sample of the Swiss population is not optimally supplied with *n-3* PUFAs.

The present study has several limitations. We tested the validity of the Swiss *n-3* PUFA FFQ compared to FR and RBC PUFA composition, but did not test reproducibility or seasonal effects. In Switzerland, fish is an episodically consumed food. It is therefore possible that during the 7-day recall period of the FR, a person consumed unproportionally high or low levels of fish compared to their general diet, while the FFQ would have assessed the frequency over a longer term (six months). This may have limited the agreement between the two methods. Nevertheless, the fact that we used seven days for the recall period and not only three or four, would again have increased the reliability. Another limitation is the different periods of our measurements in general. While the FR was completed over seven days, the FFQ assessed consumption frequencies over six months and the RBC PUFA composition represents the consumption over about 120 days. For higher precision, it might have been desirable to repeat the FR and to adapt the recall period of the FFQ to four instead of six months to match the lifetime of RBCs. Also, the sample size for the validation of the FFQ against FR was quite small with 46 participants, which limits generalizability of the results. Further, we tested the questionnaire in healthy adults and can therefore not judge its validity in other population groups. As the Swiss *n-3* PUFA FFQ, similar to any FFQ, is highly dependent on eating habits and food availability, changes in those may affect the validity of the questionnaire. Especially industry practices concerning the diet of livestock and farm-raised fish, but also economic changes and public perception can be important. Nevertheless, RBC DHA content was especially shown to be a reliable biomarker of DHA status, also reflecting a rapid reaction to an intervention similar to plasma DHA levels, which is a more short-term indicator [10]. Lastly, due to limitations in reporting, we were not able to quantify the amounts of *n-3* PUFA consumed in the form of supplements but could only correct for supplement intake as a binary variable in the regression analyses. As the number of supplement consumers was small (*n* = 14 out of 152), we do not expect this to have a major impact on the analysis. Nevertheless, when calculating *n-3* PUFA intakes, we are likely underestimating the true values to some extent. Another point worth mentioning is that for evaluation of both the FFQ and the FR we depend on data from food composition tables. Especially because the fatty acid composition of fish varies largely depending on feed, the accuracy of those tables is always limited. As the same data was used for analysis of the FFQ and the FR, we cannot rule out a correlated error to a certain extent. Nevertheless, even though this may have biased the absolute intake of *n-3* PUFA, it should not have had a huge impact on the agreement between the two methods.

In conclusion, our study shows that the adapted Swiss *n-3* PUFA FFQ is a valid tool to assess dietary *n-3* PUFA intake, especially DHA and EPA intake, and that this intake is associated with RBC PUFA status. Therefore, the Swiss *n-3* PUFA FFQ can be applied as a screening tool in population based studies, in particular to detect high risk populations with a severely insufficient *n-3* intake that should then undergo confirmatory RBC *n-3* PUFA analysis to determine if a dietary change or a supplementation with omega-3 fatty acids might be indicated.

## Figures and Tables

**Figure 1 nutrients-11-01863-f001:**
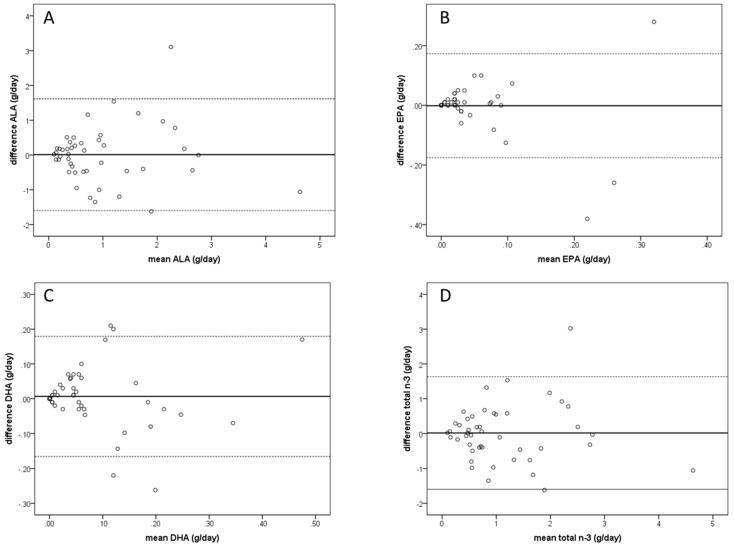
Bland Altman plots showing the agreement between 7-day food records and food frequency questionnaire to assess the intake of: (**A**) α-linolenic acid (ALA), (**B**) eicosapentanoic acid (EPA), (**C**) docosahexanoic acid (DHA), and (**D**) total *n-3* polyunsaturated fatty acid intake (total *n-3*). The limits of agreement (dotted line) indicates the 95% confidence interval (mean ±1.96 * SD).

**Table 1 nutrients-11-01863-t001:** Characteristics of the participants for the validation of an adapted food frequency questionnaire (FFQ) against 7-day weighed food records (FR) and red blood cell (RBC) fatty acid composition.

	FFQ vs. FR	FFQ vs. RBC
*n*	46	152
Gender m/f (*n* (%))	10 (22%)/36 (78%)	61 (40%)/91 (60%)
Age (year)	24 (19–53)	26 (18–59)
Height (m)	-	1.71 (1.46–1.95)
Weight (kg)	-	67.5 (36.0–109.5)
BMI (kg/m^2^)	-	22.7 (16.6–35.2)
Fish oil supplements n (%)	-	14 (9.2%)
Hormonal contraception n (%) ^1^	-	27 (29.7)

^1^ women only. BMI: body mass index.

**Table 2 nutrients-11-01863-t002:** Median *n-3* PUFA intake assessed using 7-day weighed food records (FR) and food frequency questionnaires (FFQ), correlations between *n-3* PUFA intake estimated by FR and FFQ and group comparisons of *n-3* PUFA intakes between FR and FFQ (*n* = 46).

	FR Intake		FFQ Intake		Correlations		Group Comparison ^1^
	Median (g/day)	Range	Median (g/day)	Range	Sprearman’s *r*	*p*-Value	*p*-Value
ALA	0.645	0.060–5.160	0.585	0.040–4.103	0.526	<0.001	0.915
EPA	0.010	<0.00–0.410	0.024	<0.001–0.460	0.585	<0.001	0.196
DHA	0.020	<0.001–0.390	0.050	<0.001–0.560	0.586	<0.001	0.467
Total *n-3*	0.835	0.080–5.160	0.775	0.060–4.106	0.523	<0.001	0.874

^1^ Wilcoxon signed-rank test. ALA: α-linolenic acid; EPA: eicosapentanoic acid; DHA: docosahexanoic acid.

**Table 3 nutrients-11-01863-t003:** Median *n-3* PUFA intake assessed using food frequency questionnaires (FFQ) and median RBC *n-3* PUFA composition, as well as correlations between the two (*n* = 152).

	FFQ Intake		% RBC Membrane Composition		Correlations	
	Median (g/day)	Range	Median (%)	Range	Spearman’s r	*p*-Value
ALA	0.277	0.005–6.324	0.08	0.04–0.3	0.314	<0.001
EPA	0.021	<0.001–0.348	0.53	0.26–2.12	0.430	<0.001
DHA	0.044	<0.001–0.732	5.74	1.53–9.93	0.605	<0.001
Total *n-3*	0.433	0.005–6.455	7.47	3.12–13.75	0.211	0.009
*n-3* index	-	-	6.13	1.95–12.05	-	-
*n-3*/*n*-6 ratio	-	-	4.30	2.08–11.57	-	-

**Table 4 nutrients-11-01863-t004:** Multiple linear regression models with each individual RBC *n-3* PUFA as the dependent and the respective calculated FFQ PUFA, gender, age, supplement intake (yes/no), and fish intake (yes/no) as independent variables (*n* = 152).

	Unstandardized B						Model *R*^2^
	Constant	FFQ % ^1^	Gender	Age	Supplement Intake	Fish Intake	
RBC ALA	0.102	0.013 **	−0.014 *	<0.001	−0.004	0.004	0.212
RBC EPA	0.368	1.497 **	0.016	0.005 *	0.280 **	−0.053	0.449
RBC DHA	5.344	5.029 **	−0.631 *	0.013	0.595	−1.397 **	0.430
RBC total *n-3*	7.158	0.246 *	−0.323	0.021	1.612 **	−1.874 **	0.367

^1^ For each regression, the corresponding PUFA proportions calculated from the FFQ were used (e.g., RBC ALA − FFQ ALA); * significant at *p* < 0.05; **significant at *p* < 0.001.

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
