# Peer review of "Validation of a Food Frequency Questionnaire to Assess Intake of n-3 Polyunsaturated Fatty Acids in Switzerland"

_nutrients, 2019, doi:10.3390/nu11081863_

Round 1

Reviewer 1 Report

Overall, this study provided useful information on the validation of a n-3 PUFA FFQ for use in Switzerland. A few comments to be addressed:

There may not be sufficient statistical power given the small sample size (n=46) used to validate the FFQ against 7-day FR. While statistical significance could still be observed, the small sample size should be mentioned as a limitation.  Line 165 - multinomial linear regression does not seem to be the statistical model employed. Could the authors have meant multivariable?  It would be useful to include the p-values for Bland-Altman correlation coefficients.   

Author Response

Overall, this study provided useful information on the validation of a n-3 PUFA FFQ for use in Switzerland. A few comments to be addressed:

Our reply: Thank you very much for taking the time to review our manuscript. 

There may not be sufficient statistical power given the small sample size (n=46) used to validate the FFQ against 7-day FR. While statistical significance could still be observed, the small sample size should be mentioned as a limitation. 

Our reply: We have added a comment on the small sample size in the discussion (page 10, line 374)

Line 165 - multinomial linear regression does not seem to be the statistical model employed. Could the authors have meant multivariable? 

Our reply: Thank you for noticing. It was actually a multiple regression. We have changed this in the text (page 4, line 165)

It would be useful to include the p-values for Bland-Altman correlation coefficients. 

Our reply: We are not quite sure p-values for which correlation coefficients are required. Before we did the BA plots we run one sample t-tests of the differences, which were all not significant as reported in the text. After plotting the BA plots we further run linear regressions with the difference as the dependent and the mean as the independent variables to check for proportional bias. Those results are also reported. To our understanding, the BA plot is primarily a graphical tool to visualize the agreement between two methods. The correlations between FFQ and FR were done separately and reported in Table 2 including correlation coefficients and p-values. 

Reviewer 2 Report

The aim of the paper was to adapt and validate an existing FFQ for n-3 PUFA intake from the 72 United States for the use in Switzerland using RBC n-3 fatty acid composition and 7-day food records 73 (FR) as references.  

The paper is clearly structured, contains the results of a thorough analysis and is well written.

Though there are some points of concern:

Line 205, ‘Equations 1-4 can be used to estimate RBC PUFA composition’, please explain further.

Line 206-213, for readability could you provide the B and significance (can be shown with a *) for each variable of the different equations in table form?

Some general points for inclusion in the discussion:

Fish is generally speaking an episodically consumed food, please comment on this, also regarding the validity of the use of FFQ, 7-day FR and RBC n-3 PUFA estimation. In line with this, comment on the different recall times (FFQ 6 months, RBC 120 days, FR 7 days) of the methods.

Comment on the different population groups from reference 14 and 20 compared to the population group in this study.  

To assess the n-3 PUFA intake from FFQ and FR nutrient data from food composition table was needed, please comment on the validity for n-3 PUFA content from the FCT and on expected correlated errors between FFQ and FR due to the use of the same FCT to calculate n-3 PUFA.

The number of participants in this study is small, especially for the FFQ-FR validation; and this limitation should be mentioned in the discussion section.

Author Response

The aim of the paper was to adapt and validate an existing FFQ for n-3 PUFA intake from the 72 United States for the use in Switzerland using RBC n-3 fatty acid composition and 7-day food records 73 (FR) as references.  

The paper is clearly structured, contains the results of a thorough analysis and is well written.

Our reply: Thank you very much for taking the time to comment on our manuscript. Your suggestions were highly appreciated.

Though there are some points of concern:

Line 205, ‘Equations 1-4 can be used to estimate RBC PUFA composition’, please explain further.

Our reply: We have expanded the explanation (page 5, line 199).

Line 206-213, for readability could you provide the B and significance (can be shown with a *) for each variable of the different equations in table form?

Our reply: Thank you for this suggestion. We have added Table 4 showing all the variables of the equations.

 Some general points for inclusion in the discussion:

 Fish is generally speaking an episodically consumed food, please comment on this, also regarding the validity of the use of FFQ, 7-day FR and RBC n-3 PUFA estimation. In line with this, comment on the different recall times (FFQ 6 months, RBC 120 days, FR 7 days) of the methods.

Our reply: We have added those points to the discussion of the limitations (page 10, line 363).

Comment on the different population groups from reference 14 and 20 compared to the population group in this study.

Our reply: We are already mentioning that in ref. 20 children are studies in the text (page 8, line 269) and it did not seem necessary to discuss this any further to us, as we compare the strengths of correlations between the studies and not absolute intakes. However we have introduced an additional sentence on the fact the ref 14 included not only healthy participants (page 8, line 295).

To assess the n-3 PUFA intake from FFQ and FR nutrient data from food composition table was needed, please comment on the validity for n-3 PUFA content from the FCT and on expected correlated errors between FFQ and FR due to the use of the same FCT to calculate n-3 PUFA.

Our reply: We have introduced a short discussion of this problem (page 10, line 387).

The number of participants in this study is small, especially for the FFQ-FR validation; and this limitation should be mentioned in the discussion section.

Our reply: We have added a comment on the small sample size in the discussion (page 10, line 374)